# The apical mucus layer alters the pharmacological properties of the airway epithelium

Daniela Guidone[1] ⓘ, Martina de Santis[1] ⓘ, Emanuela Pesce[2], Valeria Capurro[2], Nicoletta Pedemonte[2] ⓘ and Luis J. V. Galietta[1,3] ⓘ

[1] *Telethon Institute of Genetics and Medicine, Pozzuoli, Italy*
[2] *UOC Genetica Medica, Istituto Giannina Gaslini, Genova, Italy*
[3] *Department of Translational Medical Sciences, University of Napoli 'Federico II', Naples, Italy*

Handling Editors: Peying Fong & Péter Hegyi

The peer review history is available in the Supporting Information section of this article (https://doi.org/10.1113/JP287891#support-information-section).

**Abstract figure legend** In the airway epithelium, the cAMP-activated cystic fibrosis transmembrane conductance regulator (CFTR) channel is a major pathway for chloride secretion. Its activity can potentially be evaluated by measuring the extent of chloride flux inhibition elicited by CFTR blockers, such as CFTR$_{inh}$-172 and PPQ-102. We found that CFTR inhibitors inhibited the cAMP-activated chloride current in cultured human airway epithelia only in part. We hypothesized that the mucus layer covering the epithelium acts as a barrier that reduces the efficacy of inhibitors. Removal of mucus with the reducing agent dithiothreitol (DTT) restored CFTR inhibition.

Daniela Guidone graduated in 2017 from Federico II University in Naples, then joined Luis J.V. Galietta's laboratory at Telethon Institute of Genetics and Medicine (TIGEM) in Pozzuoli, Italy. She obtained a PhD from the European School of Molecular Medicine (SEMM) in 2023. She was recently awarded the 'Gianni Mastella Research Fellowship 2024 (GMRF#1/2024)' by the Italian Cystic Fibrosis Foundation. She is currently a postdoctoral researcher in TIGEM. Her main project concerns how human bronchial epithelia respond to inflammatory stimuli to fight bacterial infections and how these processes are impaired in cystic fibrosis.

The Journal of Physiology

**Abstract** Electrogenic transepithelial ion transport can be measured with the short-circuit current technique. Such experiments are frequently used to evaluate the activity of the cystic fibrosis transmembrane conductance regulator (CFTR), a cAMP-activated chloride channel that is defective in cystic fibrosis, one of the most frequent genetic diseases. Typically, CFTR activity is estimated from the effect of CFTR$_{inh}$-172, a selective CFTR inhibitor. Unexpectedly, we found that CFTR$_{inh}$-172, in addition to PPQ-102, another CFTR inhibitor, caused only partial inhibition of CFTR function, particularly in epithelia in pro-inflammatory conditions, which are characterized by abundant mucus secretion. We hypothesized that the mucus layer was responsible for the poor activity of CFTR inhibitors. Therefore, we treated the epithelial surface with the reducing agent dithiothreitol to remove mucus. Removal of mucus, confirmed by immunofluorescence, resulted in highly enhanced sensitivity of CFTR to pharmacological inhibition. Our results show that the mucus layer represents an important barrier whose presence limits the activity of pharmacological agents. This is particularly relevant for CFTR and the evaluation of therapeutic approaches for correction of the basic defect in cystic fibrosis.

(Received 16 October 2024; accepted after revision 11 February 2025; first published online 6 March 2025)

**Corresponding author** Luis J. V. Galietta: Telethon Institute of Genetics and Medicine (TIGEM), Via Campi Flegrei 34, 80078 Pozzuoli (NA), Italy. Email: l.galietta@tigem.it

### Key points

- Activity of the cAMP-activated cystic fibrosis transmembrane conductance regulator (CFTR) chloride channel can be evaluated by measuring the inhibition elicited by the selective blockers CFTR$_{inh}$-172 and PPQ-102.
- In short-circuit current recordings on human airway epithelia, CFTR inhibitors had only a partial effect on cAMP-dependent chloride secretion, suggesting the possible contribution of other ion channels.
- The mucus layer covering the epithelial surface was removed with the reducing agent dithiothreitol.
- Treatment of epithelia with dithiothreitol markedly improved the efficacy of CFTR inhibitors.
- The partial effect of CFTR inhibitors might be explained by the presence of the mucus layer acting as a barrier.

## Introduction

Human airways represent a first defence barrier against pathogens and noxious agents delivered with inhaled air (Knowles & Boucher, 2002). They are covered by a protective layer known as the airway surface liquid, which consists of mucus and periciliary liquid. The respiratory tract is also characterized by the presence of cilia, hair-like structures that beat in a distal-to-proximal direction and propel the mucus layer. The interaction between mucus and cilia is defined as mucociliary clearance, a coordinated and tightly regulated process that is essential to clear the airways (Vanaki et al., 2020). The crucial importance of this mechanism is demonstrated by several pathological conditions caused by mucociliary clearance dysfunction, such as cystic fibrosis (CF), primary ciliary dyskinesia, asthma and chronic obstructive pulmonary disorder (Mall, 2008).

Cystic fibrosis is one of the most frequent genetic diseases in the Caucasian population (Elborn, 2016). The disease is caused by mutations in the cystic fibrosis transmembrane conductance regulator (*CFTR*) gene, encoding for a cAMP-activated chloride-permeable channel expressed in the apical membrane of epithelial cells of various tissues and organs. The most frequent mutation is F508del, which causes both CFTR misfolding and defective channel gating (Lopes-Pacheco, 2020). For this reason, the most effective pharmacological therapy of CF patients with F508del is represented by a combination of correctors and potentiators, which are required to rescue CFTR folding and trafficking to the plasma membrane, in addition to channel activity (Keating et al., 2018; Lopes-Pacheco, 2020). In recent years, primary human bronchial and nasal epithelial cells, obtained from explanted lungs and nasal brushings, have been used as an effective *in vitro* model to evaluate the

efficacy of correctors and potentiators on mutant CFTR (Dreano et al., 2023; Keating et al., 2018; Pedemonte et al., 2020). After isolation from resected lungs or by nasal brushings, basal epithelial stem cells are expanded with a proliferative medium for a limited number of cell divisions. Then, cells are plated on porous supports and kept in air–liquid interface conditions (no medium on the apical side) to promote mucociliary differentiation (Guidone et al., 2022; Renda et al., 2023).

Differentiated airway epithelia display a relatively high electrical resistance and electrogenic ion transport that can be measured with the short-circuit current technique. In particular, the protocol to evaluate CFTR function, hence the efficacy of CFTR rescue manoeuvres, is based on maximal CFTR activation with a high concentration of a membrane-permeable cAMP analogue (e.g. CPT-cAMP) or a cAMP-elevating agent (e.g. forskolin) followed by CFTR inhibition with a selective compound, typically CFTR$_{inh}$-172 (Caci et al., 2008; Ma et al., 2002). For CF epithelia, the protocol includes the addition of a potentiator, usually VX-770 (van Goor et al., 2009), before the inhibitor. The reduction in current caused by CFTR$_{inh}$-172 is the parameter frequently used to define CFTR activity. This method has been used extensively to determine the effect of modulators on cells with a variety of CFTR mutations (Dreano et al., 2023; Pedemonte et al., 2020; Renda et al., 2023; Sondo et al., 2022; Tomati et al., 2023). However, it is not clear whether CFTR alone is responsible for cAMP-activated currents or whether other ion channels are also involved. It has been proposed that the SLC26A9 chloride channel might also contribute to cAMP-activated chloride secretion, possibly with a mechanism involving physical interaction with CFTR (Bertrand et al., 2009; Larsen et al., 2021). In this respect, Saint-Criq et al. (2020) noticed that a substantial fraction of cAMP-activated current is not inhibited by CFTR$_{inh}$-172. Also, Romano Ibarra et al. (2024) recently observed, in epithelia treated with IL-13, a large cAMP-dependent current that was only partly (~50%) inhibited by CFTR$_{inh}$-172. In CF epithelia, this large current was absent, thus indicating the involvement of CFTR. The reason for CFTR$_{inh}$-172 not being able to block CFTR function fully in non-CF epithelia was unclear.

In the present study, we investigated this intriguing issue, also considering that epithelia kept in particular differentiation media are characterized by abundant production of mucus. Our findings suggest that the mucus layer has an impact on the efficacy of pharmacological modulators added during short-circuit current recordings. Indeed, pretreatment of epithelia with reducing agents that disintegrate the mucus layer largely recovered the efficacy of CFTR$_{inh}$-172 and that of another CFTR inhibitor, PPQ-102. Our study also underscores a different behaviour of the mucus layer in pro-inflammatory conditions, with a much higher viscosity in epithelia treated with IL-17A/tumour necrosis factor-$\alpha$ (TNF-$\alpha$) in comparison to those treated with IL-4.

## Methods

### Ethical approval

Collection and use of bronchial and nasal epithelial cells were approved by the relevant ethical committee: Comitato Etico Regionale (registration numbers CER 28/2020 and ANTECER 042-09/07/2018). All donors of biological material signed an informed consent. The form used to obtain the informed consent was also approved by the ethical committee. All samples were anonymized using an alphanumerical code to protect donor identity. The studies conformed to the standards set by the latest revision of the *Declaration of Helsinki*, except for registration in a database.

### Bronchial and nasal epithelial cell expansion and differentiation

Human bronchial epithelial cells from CF and non-CF patients were provided by 'Servizio Colture Primarie' of the Italian Foundation for Cystic Fibrosis. Human nasal epithelial cells were collected as previously described (Scudieri et al., 2020; Sondo et al., 2022; Tomati et al., 2023). Expansion of basal airway stem cells (p63[+]/KRT5[+]), collected from resected lungs or by nasal brushings, was obtained with a serum-free medium previously described (Scudieri et al., 2012). To promote cell proliferation further, the medium was supplemented with bone morphogenetic protein antagonist (DMH-1, 1 μM), transforming growth factor- $\beta$ antagonist (A 83-01, 1 μM), and the rho-associated protein kinase 1 inhibitor (Y-27632, 10 μM) (Mou et al., 2016). After five or six passages, cells were seeded at high density on Snapwell (cc3801, Corning Costar), or Transwell (cc3470, Corning Costar) porous inserts. After 24 h from seeding, the proliferative medium on the basolateral side was replaced with the differentiation medium PneumaCult ALI (Stemcell Technologies). The medium on the apical side was removed to obtain the air–liquid interface (ALI) conditions. Epithelia were maintained in culture for ≥3 weeks to achieve full mucociliary differentiation. The PneumaCult ALI medium instruction manual (https://cdn.stemcell.com/media/files/pis/10000003440-PIS_03.pdf) suggests that excess mucus should be removed from the apical surface with PBS (0.5 ml per Snapwell or 0.2 ml per Transwell) at least once a week. We adapted the protocol by washing epithelia three times per week, starting from the first week of ALI culture, by

keeping PBS on the apical side for 10 min at 37°C before removal. A final wash was done 24 h before all types of experiments.

### Immunofluorescence of cultured epithelia

Snapwell supports carrying differentiated bronchial epithelial cells were fixed for 5 min in Carnoy fixative (60% absolute ethyl alcohol, 30% chloroform and 10% acetic acid, freshly prepared at the time of fixing), directly adding the fixative on the apical side. Where indicated, the epithelia were previously washed with dithiothreitol (DTT, 5 mM; 43819 Sigma-Aldrich) for 5 min at room temperature. After fixing, the epithelia were washed three times in PBS. After antigen retrieval with 10 mM citrate buffer, the samples were permeabilized with 0.3% Triton X-100 in PBS for 5 min, blocked with 1% bovine serum albumin in PBS for 2 h, then incubated overnight at 4°C with primary antibodies diluted in PBS containing 1% bovine serum albumin. The following antibodies and dilutions were used: rabbit anti-MUC5B (HPA008246, Sigma-Aldrich; 1:300) and mouse IgG1 anti-MUC5AC (MA5-12178, Thermo Fisher Scientific; 1:200). After three washes with PBS, cells were incubated with secondary fluorescent antibodies. After three further washes in PBS, slides were mounted using Fluoroshield with 4′,6-diamidino-2-phenylindole to stain cell nuclei. Images were acquired using a laser scanning confocal microscope (Nikon Eclipse Ti2E AX confocal).

### Short-circuit current recordings

Transwell supports carrying differentiated bronchial epithelia were mounted in Ussing-like vertical chambers (EM-CSYS-8, Physiologic Instruments, San Diego, CA, USA). Where indicated, the epithelia were previously washed with DTT. Both apical and basolateral chambers were filled with 5 ml of a Ringer bicarbonate solution containing (mM): 126 NaCl, 0.38 $KH_2PO_4$, 2.13 $K_2HPO_4$, 1 $CaCl_2$, 1 $MgSO_4$, 24 $NaHCO_3$, 10 glucose and Phenol Red. Solutions on both sides were bubbled with 5% $CO_2$–95% air and kept at 37°C. The transepithelial voltage was clamped at 0 mV with an eight-channel voltage-clamp amplifier (VCC MC8, Physiologic Instruments) connected to apical and basolateral chambers via Ag/AgCl electrodes and agar bridges (1 M KCl in 2% agar). The resulting short-circuit current from each channel was recorded with the Acquire & Analyse 2.3 software (Physiologic Instruments). The values of short-circuit currents in representative traces and summary graphs were normalized by epithelial surface.

### Fluorescence recovery after photobleaching assay

The assay was described in detail previously (Guidone et al., 2022). Epithelia were generated in an upside-down configuration, by seeding the cells on the bottom part of 'flipped' Snapwell inserts. After cell adhesion, Snapwell inserts were returned to their normal position, with proliferative medium on both sides. The day after, the proliferative medium was replaced with PneumaCult ALI on the top part of the insert (basolateral side with respect to the epithelium) and totally removed from the bottom part. The day of fluorescence recovery after photobleaching (FRAP) experiments, the apical surface of epithelia was stained with 5 µl PBS containing FITC-Dextran (70 kDa, 2 µg/ml, Thermo Fisher Scientific). After 3 h, Snapwells with epithelia were mounted on the stage of a Nikon Eclipse Ti-E Spinning Disk inverted microscope equipped with a chamber that allowed control of the temperature (37°C) and atmosphere (humidified 5% $CO_2$–95% air). Images of the stained epithelial surface were acquired for 5 s before and 60 s after photobleaching of a preselected circular region of interest (50 µm in diameter). After normalization for the initial value, the recovery of fluorescence at 30 s from the bleach pulse was calculated.

### Chemicals

$CFTR_{inh}$-172 (HY-16 671), PPQ-102 (HY-14 179), bumetanide (HY-17 468), VX-445 (HY-111 772) and VX-661 (HY-15 448) were from MedChemExpress. VX-770 (S1144) was from Selleck Chemicals. Ani9 (6076/10) and amiloride (0890) were from Tocris. UTP (U6625), CPT-cAMP (C3912) and all other salts and reagents were from Sigma-Aldrich.

### Statistics

Data are shown as representative traces/images or as scatter dot plots reporting the data obtained from separate experiments. Data were first analysed with the Kolmogorov–Smirnov test to evaluate normal distribution. To assess significant differences between groups of data, we used unpaired Student's *t*-test in the case of two groups or one-way ANOVA for more than two groups. ANOVA was followed by Tukey's *post hoc* test. Where indicated, if data did not pass the normality test, the Mann–Whitney *U*-test was used for twp groups and the Kruskal–Wallis test for more than two groups. Statistical analysis was done with PRISM software (GraphPad).

## Data presentation

All figures were prepared with the Igor software (Wave-Metrics).

## Results

Various research groups, including ours, presently use the PneumaCult ALI as the medium for optimal differentiation of airway epithelia. In comparison to epithelia generated with a previous home-made medium, epithelia generated with PneumaCult ALI are characterized by tall and columnar, highly ciliated cells (Scudieri et al., 2020). We also noticed a large production of mucus. During short-circuit current recordings of bronchial epithelia obtained with PneumaCult ALI, we found in many experiments that CFTR$_{inh}$-172 was less effective than expected, with only a partial inhibition of the CFTR current elicited by CPT-cAMP (Fig. 1A).

We also used another CFTR inhibitor, PPQ-102, added after CFTR$_{inh}$-172. Like CFTR$_{inh}$-172, PPQ-102 was partly effective, with persistence of a relatively large fraction of the cAMP-activated current (Fig. 1A). A similar abnormality was also observed in epithelia treated for 72 h with IL-4 (Fig. 1B). This treatment alters the ion transport properties of epithelia (Gorrieri et al., 2016). In particular, the function of the epithelial sodium channel (ENaC) is downregulated, as indicated by the reduced response to amiloride, a potent ENaC blocker. The expression and function of CFTR is instead upregulated. In agreement with CFTR upregulation, the response to CPT-cAMP was markedly increased (nearly threefold) in epithelia treated with IL-4 (compare Fig. 1A and B). However, CFTR$_{inh}$-172 was still partly active, with a large residual current, also refractory to PPQ-102. Graphs in Fig. 1 also depict a high degree of variability, with experiments in which the inhibition by CFTR$_{inh}$-172 exceeded 50% and others in which the effect was much smaller. The kinetics of CFTR inhibition were also variable, with cases in which current rundown was slow.

We wondered whether the abundant mucus production of bronchial epithelial cells differentiated in Pneumacult

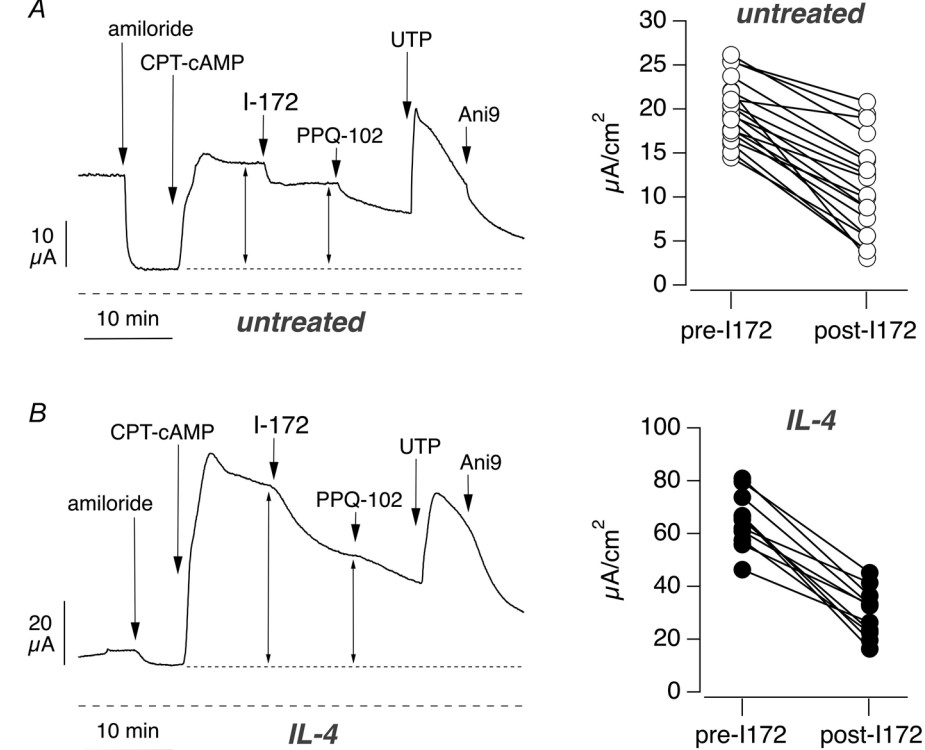

**Figure 1. Efficacy of CFTR inhibitors in airway epithelia**
Representative traces (left) and summary graphs (right) showing short-circuit current results obtained in epithelia untreated (A) or treated (B) with IL-4 (10 ng/ml) for 72 h. During short-circuit current recordings, epithelia were exposed sequentially to: amiloride (10 μM, apical), CPT-cAMP (100 μM, apical and basolateral), CFTR$_{inh}$-172 (I-172, 10 μM, apical), PPQ-102 (30 μM, apical), UTP (100 μM, apical) and Ani9 (10 μM, apical). Dotted line indicates the baseline that was used to measure cAMP-activated current before and after I-172. Dashed line indicates absolute zero current level. The graphs report, for each experiment, the value of the cAMP-activated current before and after CFTR$_{inh}$-172 addition. Data in the figure (n = 11–19 per condition) are from separate preparations of cells from three non-cystic fibrosis donors.

ALI hampers the effect of inhibitors during short-circuit current recordings. To detect mucus on the airway surface, we carried out immunofluorescence by directly adding the fixative on the surface of epithelia, without washing, and using antibodies for MUC5B and MUC5AC. In parallel, we performed immunofluorescence on epithelia prewashed for 5 min with DTT (5 mM) as a reducing agent to dissolve mucus. We used Carnoy, a non-aqueous fixative known to preserve mucus.

The images in Fig. 2 show that in Carnoy-fixed epithelia without washing there is an abundance of MUC5B filaments that form large bundles on the apical epithelial surface. Filaments of MUC5AC are also visible, particularly in epithelia treated with IL-4 (Fig. 2). Below the mucin filaments, it is possible to notice individual spots of MUC5B and MUC5AC that we interpret as single cells loaded with mucins. Importantly, the treatment with DTT completely removed the layer of mucin filaments and bundles, unveiling a large number of spots strongly positive for MUC5AC and MUC5B (Fig. 2).

We carried out short-circuit current experiments to investigate the effect of DTT washing on the extent of CFTR inhibition. As shown by representative traces and scatter dot plots in Fig. 3, epithelia washed with DTT displayed a larger reduction in current after addition of $CFTR_{inh}$-172, with near full inhibition of the cAMP-activated current.

An interesting observation concerns the behaviour of epithelia treated with/without IL-4. In epithelia without IL-4 and no DTT wash (Fig. 3A), $CFTR_{inh}$-172 appeared largely ineffective, with only 20% of the cAMP-activated current being blocked. In IL-4-treated epithelia with no DTT washing (Fig. 3B), the inhibitor was significantly more effective, with >50% inhibition. However, in both conditions, DTT washing was able to increase $CFTR_{inh}$-172 efficacy significantly (Fig. 3A, B).

Another type of pro-inflammatory stimulus with a profound effect on airway surface properties is treatment with IL-17A/TNF-$\alpha$ (Guidone et al., 2022). We evaluated the efficacy of $CFTR_{inh}$-172 in these conditions, with/without DTT washing. Without the washing, the inhibitor was largely ineffective, with results similar to epithelia without cytokine treatment. With washing, inhibitor efficacy was significantly enhanced (Fig. 4A).

We also investigated the expression of MUC5B/MUC5AC in epithelia treated with IL-17A/TNF-$\alpha$. Without DTT washing, we noticed a strong MUC5B signal (Fig. 4B), in agreement with expression of this mucin being upregulated by the cytokine combination (Guidone et al., 2022). Interestingly, the MUC5B signal was not organized as filaments or bundles. This might depend on the reduced mucociliary transport that is observed in epithelia treated with IL-17A/TNF-$\alpha$ (Guidone et al., 2022). After washing the apical surface

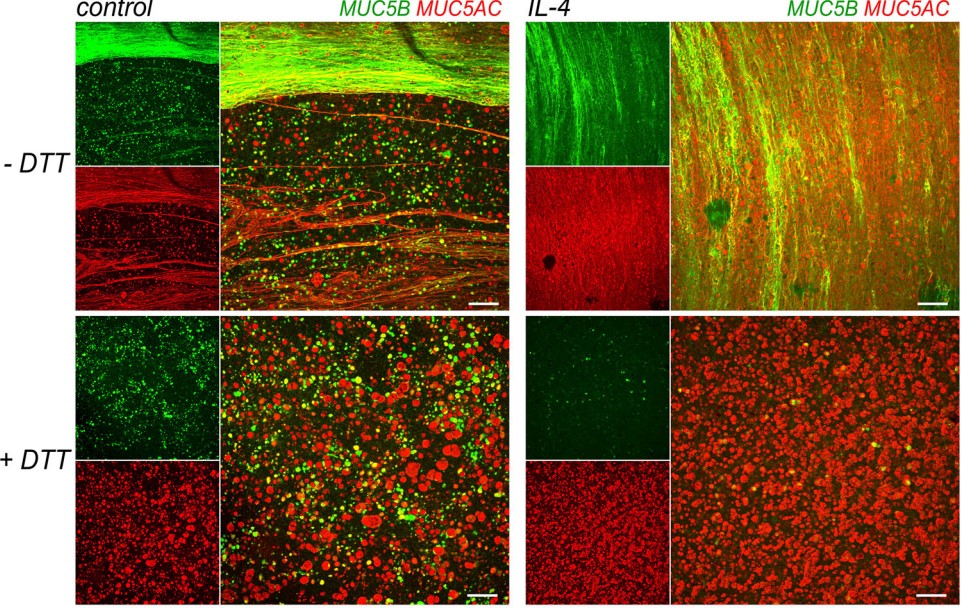

**Figure 2. Removal of mucus layer by dithiothreitol**
Detection of mucins MUC5B and MUC5AC by immunofluorescence in cultured bronchial epithelia kept in control conditions (left) or treated for 72 h with 10 ng/ml IL-4 (right). The epithelial surface was left unwashed (top) or washed with saline containing 5 mM dithiothreitol (DTT) as a reducing agent (bottom). In each panel, the small images show separate detection of MUC5B or MUC5AC. The large image reports the merge of the two mucin signals. Scale bars: 100 μm.

with DTT, the signal for MUC5B and MUC5AC was substantially reduced (Fig. 4*B*).

We asked whether the efficacy of PPQ-102, as another inhibitor of CFTR, was also improved by DTT treatment. Figure 5 shows that the reducing agent significantly increased the block of cAMP-activated current by PPQ-102.

We also used bumetanide, a blocker of the basolateral $Na^+/K^+/2Cl^-$ cotransporter (SLC12A2), to assess the contribution of $Cl^-$ secretion to the current remaining after addition of the CFTR inhibitor (Fig. 6). Given that it is added from the basolateral side, bumetanide should not be affected by mucus. We reasoned that, by acting on the same process, i.e. CFTR-dependent $Cl^-$

secretion, an increase in CFTR inhibitor efficacy induced by DTT should be paralleled by a decrease in the effect of bumetanide. Indeed, after DTT treatment, the fraction of the cAMP-dependent current blocked by bumetanide decreased and that blocked by $CFTR_{inh}$-172 increased (Fig. 6). Interestingly, the total effect of $CFTR_{inh}$-172 and of bumetanide was close to full inhibition of the cAMP-activated current in epithelia kept in control conditions or treated with IL-4 (Fig. 6*A* and *B*). Instead, a substantial fraction of the current remained unblocked in epithelia treated with IL-17A/TNF-$\alpha$ (Fig. 6*C*).

Dithiothreitol is a strong reducing agent that could affect various epithelial processes. Therefore, we evaluated the effect DTT on ENaC, CFTR and TMEM16A electro-

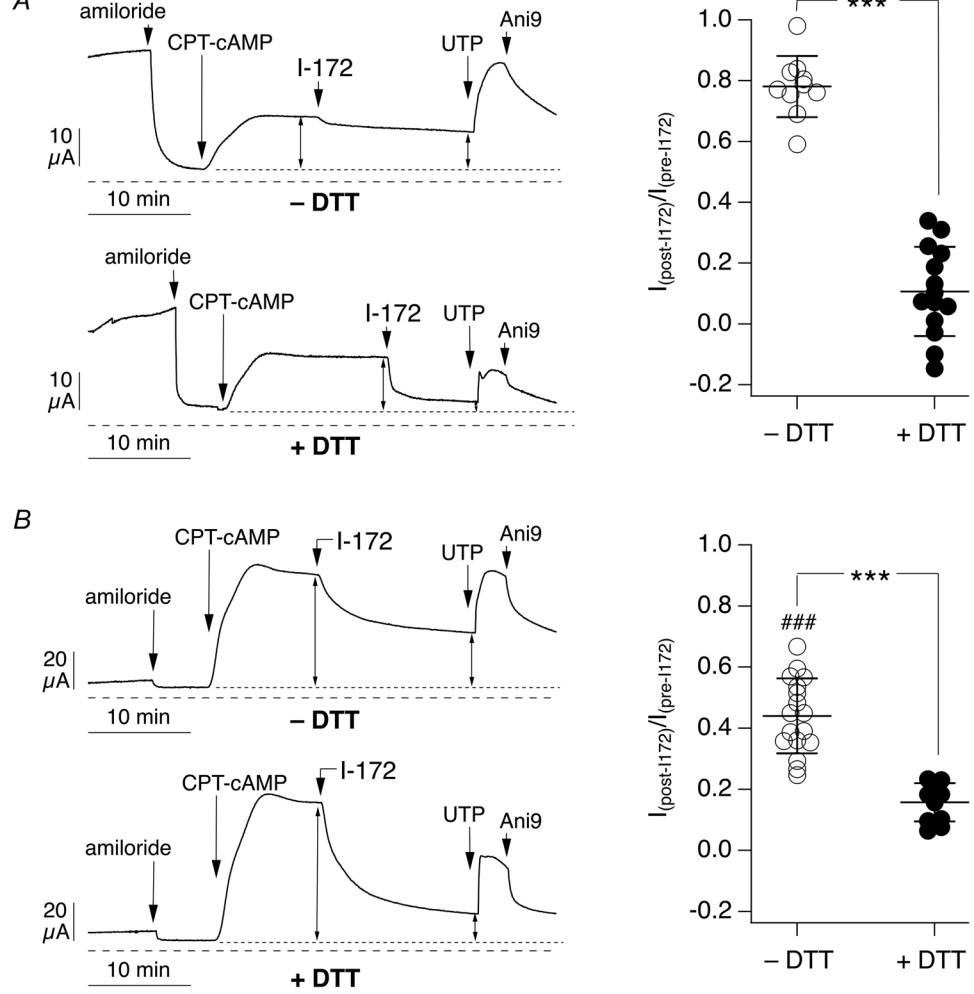

**Figure 3. Effect of mucus removal on CFTR$_{inh}$-172 efficacy**
Representative traces (left) and summary graphs (right) showing short-circuit current results obtained in epithelia untreated (*A*) or treated (*B*) with IL-4 (10 ng/ml) for 72 h. Before experiments, the apical surface of epithelia was left unwashed or washed with dithiothreitol (DTT, 5 mM). Compounds added during short-circuit current recordings were as listed for Fig. 1 except for PPQ-102, which was omitted. Dotted line indicates the baseline used to measure cAMP-activated current before and after I-172. Dashed line indicates the absolute zero current level. The graphs report the fraction of the cAMP-activated current that remains after addition of CFTR$_{inh}$-172. ***$P < 0.001$ *vs*. no wash; ###$P < 0.001$ IL-4 no wash *vs*. untreated no wash (ANOVA with Tukey's *post hoc* test). Data shown (*n* = 10–17 per condition) are from separate preparations of cells from three non-cystic fibrosis donors.

genic activity by measuring the effects of amiloride, CPT-cAMP and UTP, respectively. Figure 7 shows that most processes are unaffected, with the exception of epithelia treated with IL-17A/TNF-$\alpha$, in which ENaC activity was modestly but significantly increased by DTT.

Previously, we used the FRAP technique to investigate the effect of IL-17A/TNF-$\alpha$ (Guidone et al., 2022). In the present study, we carried out FRAP experiments also including the treatment with IL-4. Representative traces and scatter dot plots (Fig. 8) show that after photobleaching, the fluorescence in epithelia treated with IL-17A/TNF-$\alpha$ underwent a very slow recovery, which is indicative of a high viscosity of the epithelial surface, as

previously reported (Guidone et al., 2022). In contrast, recovery of fluorescence in epithelia treated with IL-4 was only slightly slowed down in comparison to untreated epithelia (Fig. 8).

The extent of the CFTR$_{inh}$-172 inhibitory effect is normally used to estimate the efficacy of pharmacological and genetic manoeuvres to rescue the function of mutant CFTR (Amistadi et al., 2023; Dreano et al., 2023; Pedemonte et al., 2020). We carried out experiments on bronchial epithelial cells from a CF patient with an F508del/F508del genotype. To target the trafficking/folding defect caused by the mutation, we treated epithelia with a combination of pharmacological

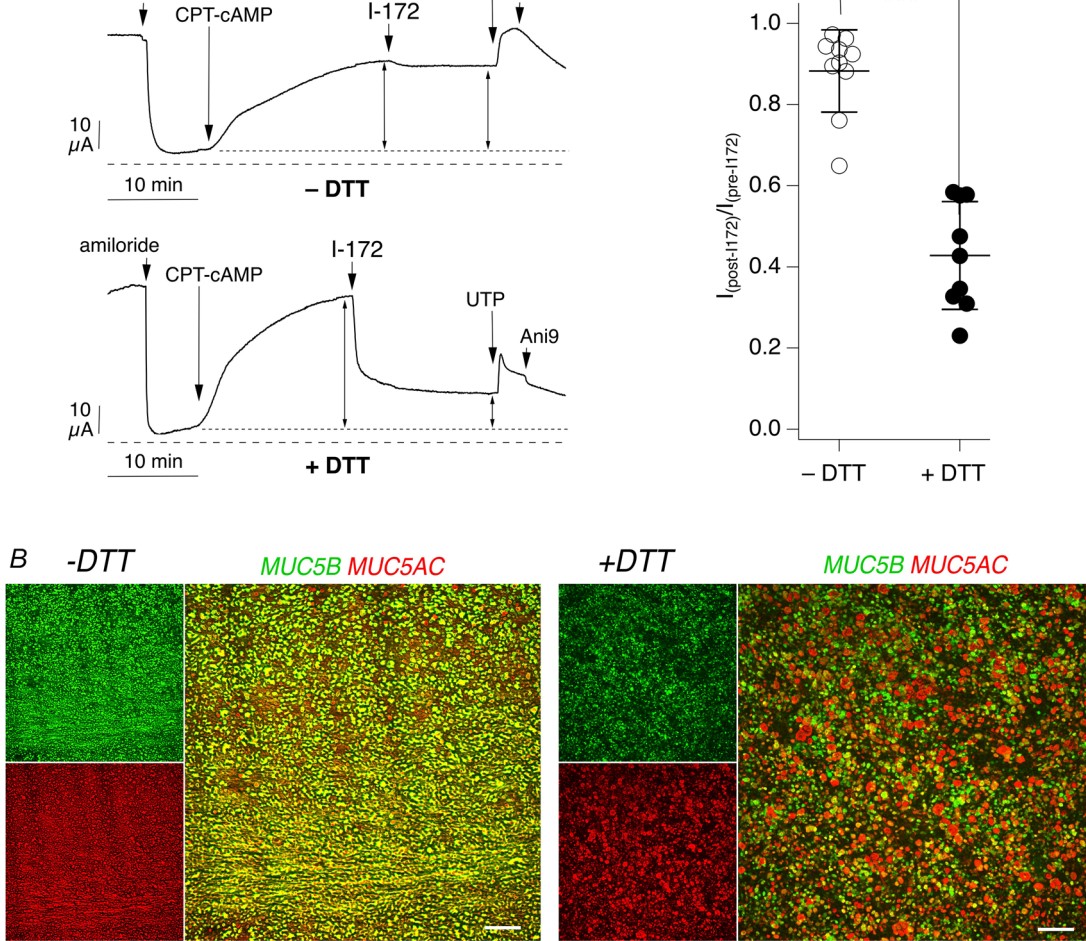

**Figure 4. Efficacy of CFTR$_{inh}$-172 in epithelia treated with IL-17A/TNF-$\alpha$**

*A*, representative traces (left) and summary graph (right) showing short-circuit current results obtained in epithelia treated with IL-17A (20 ng/ml) plus TNF-$\alpha$ (10 ng/ml) for 72 h. Before experiments, the apical surface of epithelia was left unwashed or washed with dithiothreitol (DTT, 5 mM). Dotted and dashed lines indicate the baseline for cAMP-activated current and zero current level, as in previous legends. The graph reports the fraction of the cAMP-activated current that remains after addition of CFTR$_{inh}$-172. ***$P < 0.001$ *vs.* no wash (Mann–Whitney *U*-test). Data ($n$ = 9 or 10 per condition) are from separate preparations of cells from two non-CF donors. *B*, representative immunofluorescence images showing detection of MUC5B and MUC5AC mucins in epithelia treated with the IL-17A/TNF-$\alpha$ combination and left unwashed (left) or washed (right) with DTT (5 mM). Small images show separate detection of MUC5B or MUC5AC. The large images show the merge of the two mucin signals. Scale bar: 100 µm.

correctors, namely VX-661 and VX-445. These correctors are presently used to treat patients with one or two F508del alleles (Keating et al., 2018). As shown in Fig. 9*A* and *B*, treatment with correctors for 24 h led to enhanced CFTR function, as indicated by the response to the cAMP analogue and to the CFTR potentiator VX-770, which targets the F508del-dependent channel gating defect (van Goor et al., 2009). Importantly, with DTT wash, CFTR$_{inh}$-172 fully inhibited the current activated by cAMP and the potentiator. Without DTT washing, CFTR$_{inh}$-172 was partly effective, thus leading to a potential underestimation of mutant CFTR rescue.

We also studied nasal epithelia generated *in vitro* from CF patients with orphan mutations. Nasal epithelia are widely used to define the theratype of patients, i.e. the

sensitivity to CFTR pharmacological modulators (Dreano et al., 2023; Sondo et al., 2022; Tomati et al., 2023). The results allow the identification of patients with rare mutations who can benefit from the CFTR modulators that are already approved for the treatment of patients with more frequent mutations, such as F508del. Figure 9*C* and *D* shows data obtained from two patients with the N1303K/R1066C (donor ID: GE006) and G542X/R1066C (donor ID: GE001) genotypes. Epithelia were treated with/without 3 µM VX-445 (elexacaftor, ELX) as corrector for 24 h. Before short-circuit current recordings, epithelia were washed with/without DTT. Epithelia that were washed with DTT showed a significantly larger effect of CFTR$_{inh}$-172, thus revealing a higher rescue of mutant CFTR by the corrector.

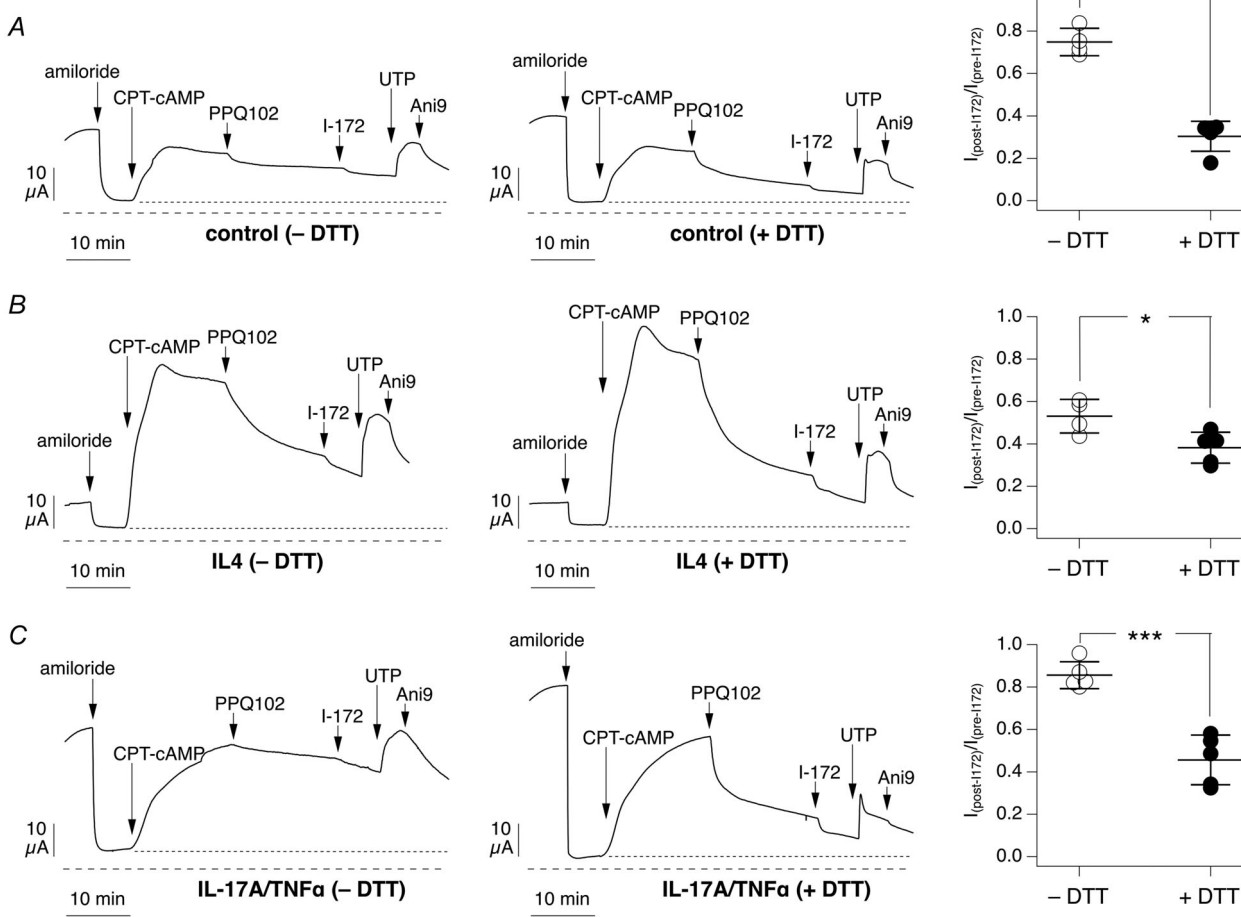

**Figure 5. Effect of mucus removal on PPQ-102 efficacy**
Representative short-circuit current traces (left) and summary graphs (right) from epithelia kept in control conditions (*A*), treated with IL-4 (*B*) or treated with IL-17A/TNF-$\alpha$ (*C*). Where indicated, epithelia were washed apically with dithiothreitol (DTT) to remove mucus. During recordings, after CFTR stimulation with CPT-cAMP, currents were inhibited with apical PPQ-102 (30 µM) followed by apical CFTR$_{inh}$-172 (I-172, 10 µM). Epithelia were then stimulated with apical UTP (100 µM) followed by apical Ani9 (10 µM). Dotted and dashed lines indicate the baseline for cAMP-activated current and zero current level, as in previous legends. The graphs report the fraction of the cAMP-activated current that remains after addition of PPQ-102, with/without DTT treatment. $*P = 0.0220$; $***P < 0.001$ (unpaired Student's *t*-test). Data ($n = 4$ or 5 per condition) are from cells of one non-CF donor.

## Discussion

Our study reveals an important role of the mucus covering airway epithelia as a barrier limiting the access of small molecules to target cells. In particular, we found that CFTR$_{inh}$-172 and PPQ-102, widely used to evaluate the contribution of CFTR to transepithelial ion transport, have a low efficacy in inhibiting cAMP-activated chloride secretion in short-circuit current recordings. Efficacy of both compounds is greatly improved by previously washing the apical surface of epithelia with a DTT-containing solution in order to remove mucus. Immunofluorescence analysis demonstrates that this washing step is indeed effective in removing the layer of mucin filaments and bundles. Interestingly, the mucus layer was not a significant barrier for two other small molecules, amiloride and UTP, which were also added during short-circuit current recordings on the apical side of epithelia. This type of result suggests that it is not the size of the molecule that determines its ability to penetrate the mucus layer but the specific physical/chemical properties that influence its interaction with mucus. In this respect, it should be taken into account that amiloride and UTP are hydrophilic substances, with concentrated stocks being prepared in aqueous solutions, whereas CFTR inhibitors require DMSO as a solvent given their more hydrophobic properties. The size might be of greater importance for large molecules, such as antibodies. Indeed, we noticed that without DTT washing, immunostaining of mucins inside cells was less effective.

Another interesting type of finding in our experiments concernss the behaviour of CFTR inhibitors on epithelia previously treated with pro-inflammatory stimuli. IL-4, like IL-13, is a Th2 cytokine that induces mucus cell metaplasia, i.e. a large increase in the number of goblet cells with overexpression of mucins, particularly MUC5AC (Gorrieri et al., 2016). IL-4 has also a profound effect on transepithelial ion transport, with suppression of ENaC function and upregulation of chloride secretion mediated by CFTR and TMEM16A (Gorrieri et al., 2016). IL-17A is instead a cytokine that, particularly in combination with TNF-$\alpha$, promotes ENaC function

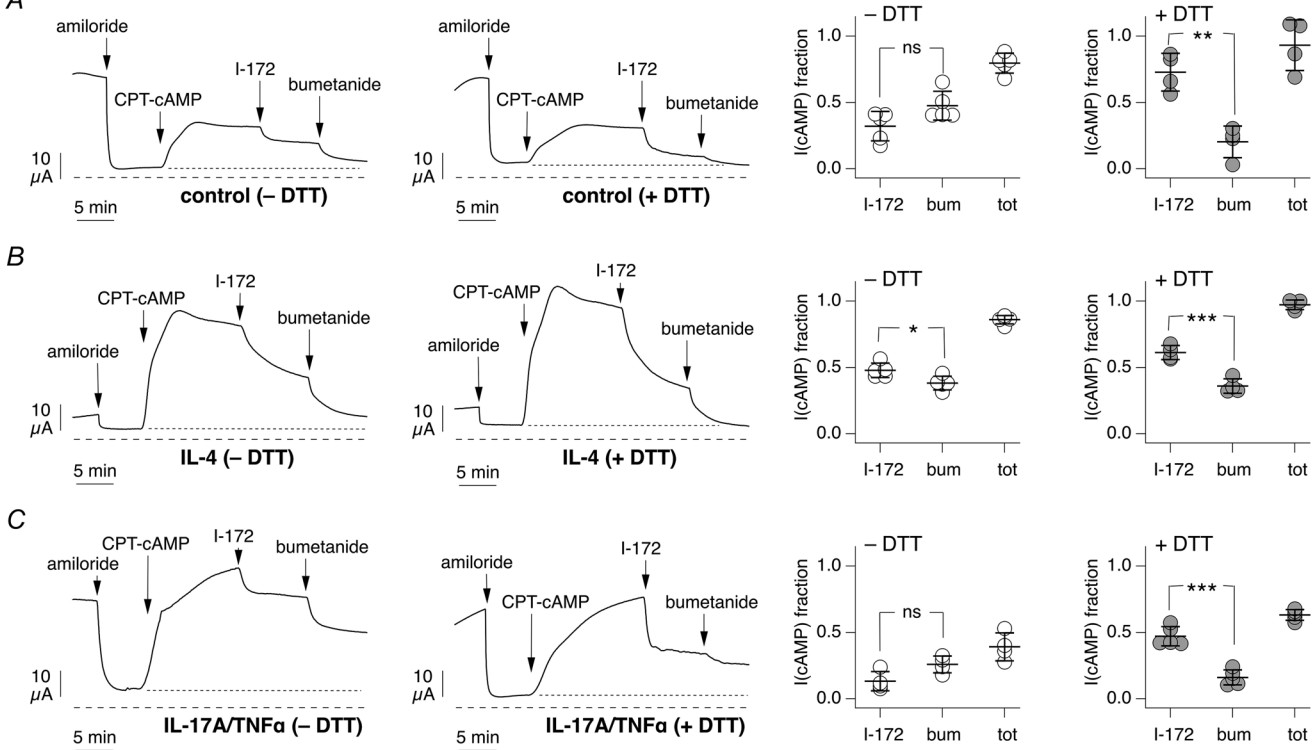

**Figure 6. Effect of bumetanide after CFTR$_{inh}$-172 addition**
Representative short-circuit current traces (left) and summary graphs (right) from epithelia kept in control conditions (*A*), treated with IL-4 (*B*) or treated with IL-17A/TNF-$\alpha$ (*C*). During recordings, CFTR$_{inh}$-172 (apical, 10 μM) and bumetanide (basolateral, 100 μM) were added sequentially after stimulation of CFTR with CPT-cAMP. Dotted and dashed lines indicate the baseline for cAMP-activated current and zero current level, as in previous legends. The graphs report the fraction of the cAMP-activated current blocked by CFTR$_{inh}$-172 (I-172), bumetanide alone (bum) or both compounds together (tot). Symbols in *A*: ns, not significant; **$P = 0.00240$. Symbols in *B*: *$P = 0.0177$; ***$P < 0.001$. Symbols in *C*: ns, not significant; ***$P < 0.001$ (ANOVA with Tukey's *post hoc* test). Data ($n = 4$ or 5 per condition) are from cells of one non-CF donor.

with dehydration and enhanced viscosity of the epithelial surface (Guidone et al., 2022). IL-17A/TNF-$\alpha$ also upregulates expression of MUC5B mucin. We were surprised to find that epithelia treated with IL-4, in the absence of DTT washing, were more responsive to CFTR$_{inh}$-172 in comparison to untreated epithelia

or epithelia treated with IL-17A/TNF-$\alpha$. This finding would suggest that, despite abundant mucus production, the apical surface of epithelia treated with IL-4 is relatively permeable to CFTR inhibitors. This condition could result from enhanced hydration of mucus owing to enhanced chloride secretion and reduced sodium absorption. To investigate this issue further, we performed FRAP experiments to estimate the viscosity of epithelia in various conditions. As reported previously (Guidone et al., 2022), epithelia treated with IL-17A/TNF-$\alpha$ showed a high viscosity in comparison to untreated epithelia. In contrast, the surface of epithelia treated with IL-4 appeared fluid. This could explain the better accessibility of CFTR inhibitors to the apical membrane.

As an additional test to investigate the basis of the residual current remaining after CFTR inhibition, we have carried out experiments with bumetanide, which blocks the basolateral Na$^+$/K$^+$/2Cl$^-$ cotransporter. Interestingly, the relative effects of CFTR$_{inh}$-172 and bumetanide are inversely related. In DTT-washed epithelia, the relative effect of CFTR$_{inh}$-172 was increased and that of bumetanide was decreased, as expected if the two agents act on the same pathway (i.e. CFTR-dependent chloride transport). Also, the total effect of CFTR$_{inh}$-172 plus bumetanide was close to full inhibition of the cAMP-activated current in epithelia kept in control conditions or treated with IL-4. Instead, a substantial fraction of the current remained unblocked in epithelia treated with IL-17A/TNF-$\alpha$. A possible explanation is that, besides chloride secretion, these epithelia have an increased CFTR-mediated bicarbonate secretion, as

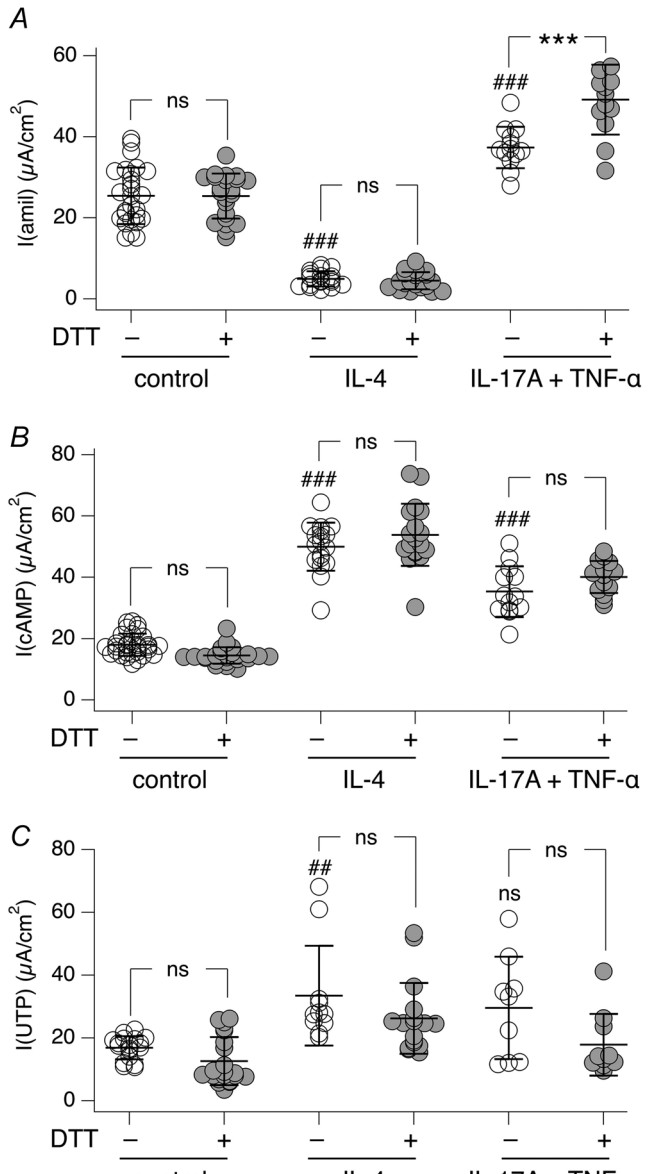

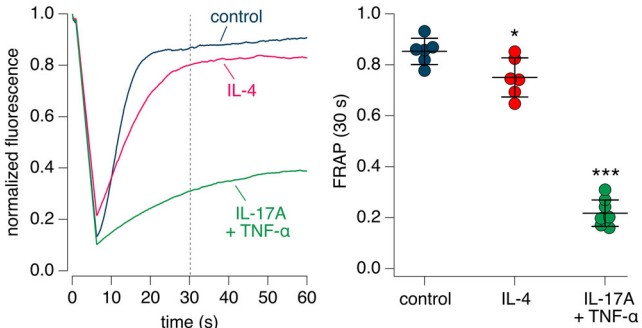

**Figure 8. Analysis of apical viscosity by fluorescence recovery after photobleaching**
*A*, representative traces showing the time course of fluorescence recovery after photo-bleaching (FRAP). Epithelia were kept in control conditions or treated for 72 h with IL-4 or with IL-17A/TNF-$\alpha$. The vertical dotted line shows the time (30 s) at which the value of fluorescence was taken. *B*, graph reporting the fluorescence recovery at 30 s post-bleaching for epithelia kept in control conditions or treated with cytokines. *$P = 0.0241$; ***$P < 0.001$ *vs*. control (ANOVA with Tukey's *post hoc* test). Data in the figure ($n = 6$–8 per condition) are from separate preparations of cells from two non-cystic fibrosis donors. [Colour figure can be viewed at wileyonlinelibrary.com]

**Figure 7. Effect of dithiothreitol treatment on electrogenic epithelial transport systems**
Graphs show the amplitude of the current blocked by amiloride (*A*), activated by cAMP (*B*) or elicited by UTP (*C*). Epithelia were kept for 72 h in control conditions or treated with cytokines as indicated, with/without dithiothreitol (DTT) treatment. Symbols in *A* and *B*: ns, not significant; ***$P < 0.001$ *vs*. no DTT; ###$P < 0.001$ *vs*. control without DTT (ANOVA with Tukey's *post hoc* test). Symbols in *C*: ns, not significant; ##$P = 0.00850$ *vs*. control without DTT (Kruskal–Wallis test). Data shown ($n = 9$–31 per condition) are from separate preparations of cells from three non-cystic fibrosis donors.

already reported by others (Zajac et al., 2023). In such a case, the bicarbonate transport would occur through another basolateral transporter, insensitive to bumetanide.

In conclusion, we found that the ability of inhibitors to block CFTR-dependent anion transport is influenced by the mucus layer covering the apical surface of airway epithelia. Apical washing with a reducing agent appears as a possible step in the preparation of epithelia for short-circuit experiments. However, DTT is a strong reducing agent that could alter different biological processes. Our experiments have shown near negligible

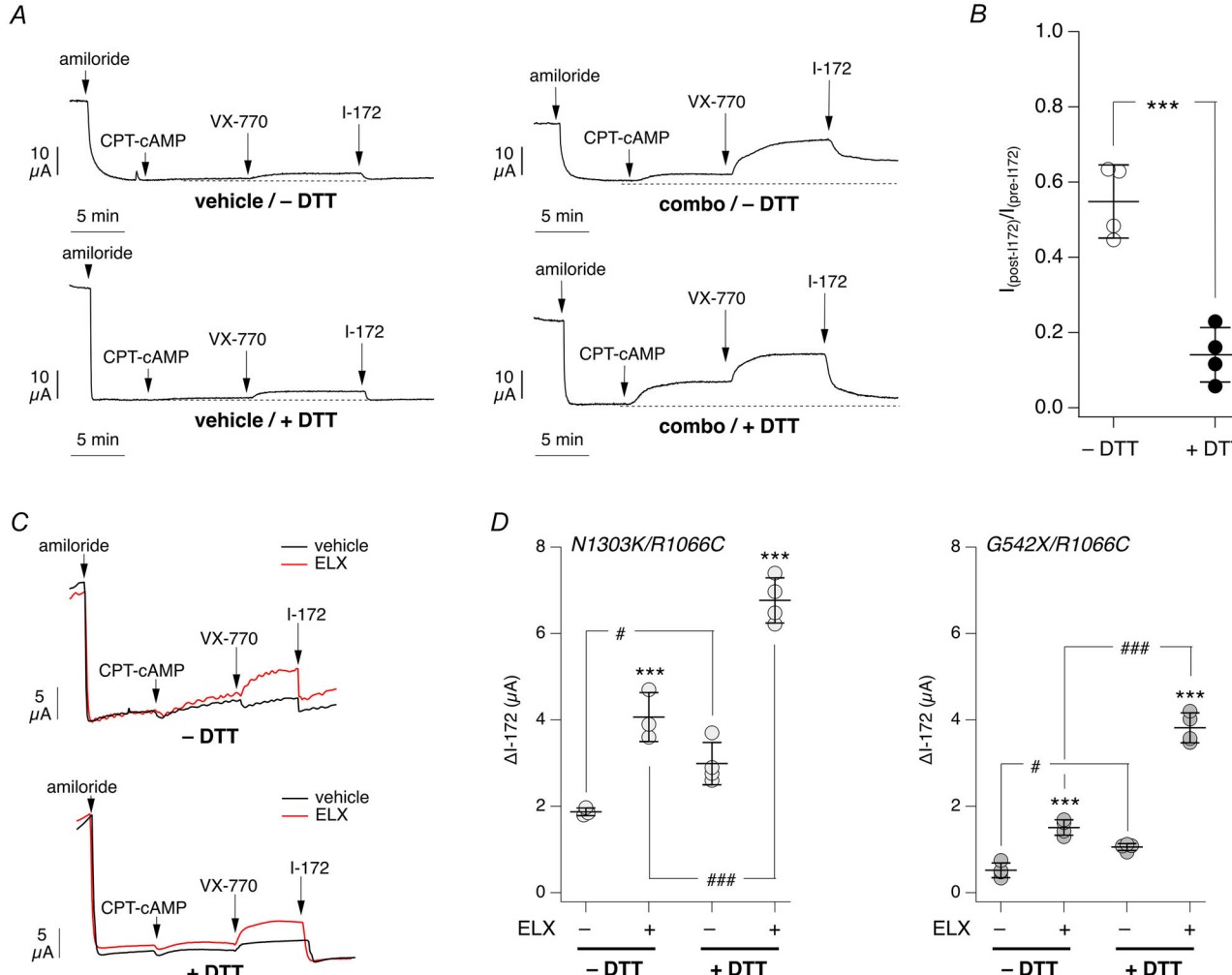

**Figure 9. Evaluation of mutant CFTR rescue**
*A*, representative short-circuit recordings on epithelia generated from a F508del/F508del cystic fibrosis patient. Epithelia were treated for 24 h with DMSO (vehicle) or with 10 μM VX-661 plus 5 μM VX-445 (combo). Before recordings, epithelia were kept unwashed (top) or washed (bottom) with dithiothreitol (DTT, 5 mM). During recordings, CFTR function was stimulated maximally with CPT-cAMP (100 μM, apical and basolateral) followed by VX-770 (1 μM, apical), before inhibition with $CFTR_{inh}$-172. *B*, graph reporting the fraction of the CFTR current (activated by CPT-cAMP plus VX-770) that remains after addition of $CFTR_{inh}$-172 in epithelia treated with correctors, with/without DTT wash. ****P* < 0.001 *vs*. no wash (unpaired Student's *t*-test). Data (*n* = 4 per condition) are from cells of one CF donor. *C*, representative short-circuit current recordings from nasal epithelia derived from a cystic fibrosis patient (donor ID: GE006) with the N1303K/R1066C genotype. Epithelia were treated with vehicle or 3 μM VX-445 (ELX) for 24 h. Before experiments, the apical surface of epithelia was washed without (top) or with (bottom) DTT. *D*, summary of results obtained in nasal epithelia derived from patients with N1303K/R1066C or G542X/R1066C genotype (donor IDs: GE006 and GE001, respectively). Data report the amplitude of $CFTR_{inh}$-172 effect for the indicated conditions. ****P* < 0.001 for ELX-treated *vs*. vehicle-treated epithelia. #*P* = 0.0457 and 0.0196 for DTT-treated vs. DTT-untreated epithelia (N1303K/R1066C and G542X/R1066C, respectively); ###*P* < 0.001 for DTT-treated *vs*. DTT-untreated epithelia (for both N1303K/R1066C and G542X/R1066C; ANOVA with Tukey's *post hoc* test). Data (*n* = 3 or 4 per condition) are from cells of each cystic fibrosis donor. [Colour figure can be viewed at wileyonlinelibrary.com]

effects on amiloride-sensitive sodium absorption, cAMP-activated current and calcium-activated chloride secretion. However, effects on other mechanisms cannot be excluded and should be checked in control experiments. Alternatively, other washing procedures, excluding DTT, could be investigated to remove mucus. In general, mucus removal appears to be an important step to record maximal CFTR activity, particularly in studies on CF epithelia evaluating the efficacy of pharmacological interventions to rescue CFTR function.

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

## Additional information

### Data availability statement

All data in this study are included in the present manuscript.

## Competing interests

Authors have no competing interests to disclose.

## Author contributions

D.G., N.P. and L.J.V.G. designed the study; D.G., M.D.S., E.P. and V.C. acquired the data; D.G., N.P. and L.J.V.G. analysed the data; D.G. and L.J.V.G. wrote the initial draft of the manuscript. All authors critically read and approved the final version of the manuscript and agree to be accountable for all aspects of the work in ensuring that questions related to the accuracy or integrity of any part of the work are appropriately investigated and resolved. All persons designated as authors qualify for authorship, and all those who qualify for authorship are listed.

## Funding

This work was supported by grants from the Cystic Fibrosis Foundation (GALIET22I0), the Italian Cystic Fibrosis Foundation (FFC#9/2022 and GMRF#1/2024) and the Italian Ministry of Health (GR-2018-12 367 126).

## Acknowledgements

The authors thank Roman Polishchuk and Roberta Crispino (Advanced Microscopy Facility, TIGEM) for their technical support.

Open access publishing facilitated by Universita degli Studi di Napoli Federico II, as part of the Wiley - CRUI-CARE agreement.

## Keywords

airway epithelium, cystic fibrosis transmembrane conductance regulator, chloride channel, channel blocker, mucus

## Supporting information

Additional supporting information can be found online in the Supporting Information section at the end of the HTML view of the article. Supporting information files available:

**Peer Review History**

