## [Peer Review History · The Journal of Physiology]

THE APICAL MUCUS LAYER ALTERS THE PHARMACOLOGICAL PROPERTIES OF THE AIRWAY EPITHELIUM

Daniela Guidone, Martina De Santis, Emanuela Pesce, Valeria Capurro, Nicoletta Pedemonte, and Luis J.V. Galietta
DOI: 10.1113/JP287891

Corresponding author(s): Luis Galietta (l.galietta@tigem.it)

The following individual(s) involved in review of this submission have agreed to reveal their identity: Shmuel Muallem (Referee #2)

Review Timeline:

Submission Date:	16-Oct-2024
Editorial Decision:	12-Nov-2024
Revision Received:	04-Jan-2025
Editorial Decision:	27-Jan-2025
Revision Received:	27-Jan-2025
Accepted:	11-Feb-2025

Senior Editor: Peking Fong

Reviewing Editor: Péter Hegyi

Transaction Report:

Dear Dr Galietta,

Re: JP-RP-2024-287891 "THE APICAL MUCUS LAYER ALTERS THE PHARMACOLOGICAL PROPERTIES OF THE AIRWAY EPITHELIUM" by Daniela Guidone, Emanuela Pesce, Valeria Capurro, Nicoletta Pedemonte, and Luis J.V. Galietta

Thank you for submitting your manuscript to The Journal of Physiology. It has been assessed by a Reviewing Editor and by 2 expert referees and we are pleased to tell you that it is acceptable for publication following satisfactory revision.

REVISION CHECKLIST:

We look forward to receiving your revised submission.

Yours sincerely,

Peying Fong
Senior Editor
The Journal of Physiology

REQUIRED ITEMS

- Your manuscript must include a complete Additional Information section, including competing interests; funding; author contributions and acknowledgements.
- Please upload separate high-quality figure files via the submission form.
- A Data Availability Statement is required for all papers reporting original data. This must be in the Additional Information section of the manuscript itself. It must have the paragraph heading 'Data Availability Statement'. All data supporting the results in the paper must be either: in the paper itself; uploaded as Supporting Information for Online Publication; or archived in an appropriate public repository. The statement needs to describe the availability or the absence of shared data. Authors must include in their statement: a link to the repository they have used, or a statement that it is available as Supporting Information; reference the data in the appropriate sections(s) of their manuscript; and cite the data they have shared in the References section. Whenever possible, the scripts and other artefacts used to generate the analyses presented in the paper should also be publicly archived. If sharing data compromises ethical standards or legal requirements then authors are not expected to share it, but must note this in their statement. For more information, see our Statistics Policy.
- Please include an Abstract Figure file, as well as the Figure Legend text within the main article file. The Abstract Figure is a piece of artwork designed to give readers an immediate understanding of the research and should summarise the main conclusions. If possible, the image should be easily 'readable' from left to right or top to bottom. It should show the physiological relevance of the manuscript so readers can assess the importance and content of its findings. Abstract Figures should not merely recapitulate other figures in the manuscript. Please try to keep the diagram as simple as possible and without superfluous information that may distract from the main conclusion(s). Abstract Figures must be provided by authors no later than the revised manuscript stage and should be uploaded as a separate file during online submission labelled as File Type 'Abstract Figure'. Please also ensure that you include the figure legend in the main article file. All Abstract Figures should be created using BioRender. Authors should use The Journal's premium BioRender account to export high-resolution images. Details on how to use and access the premium account are included as part of this email.

EDITOR COMMENTS

Reviewing Editor:

Comments to the Author:

The findings show that the mucus layer reduces CFTRinh-172 efficacy, which may lead to underestimation of CFTR-mediated ion transport in experimental models if the layer is not removed. Given that CFTR inhibitors are widely used in electrophysiological studies for evaluating personalized CF treatment strategies, this is an important factor to consider. To enhance the study's robustness the authors should

- 1) Include data on residual currents after maximal cAMP stimulation and CFTR inhibition, potentially with bumetanide, to clarify whether alternative chloride transport pathways contribute to the observed currents.

2) Discuss how more frequent PBS washes during differentiation, as done in some laboratories, might impact mucus accumulation and CFTRinh-172 efficacy.

3) Evaluate DTT Effects

4) Add clarifications to Figures 1, 2, and 4, specifying what each data point represents (e.g., individual filters or donors) and the number of technical and biological replicates.

5) Test the impact of mucus stripping on the efficacy of other inhibitors like PPQ-102

Senior Editor:

Comments for Authors to ensure the paper complies with the Statistics Policy:

Please ensure complete compliance with Statistical Policy. There are some data shown on figures 5 and 6 that are reported with p values < 0.05. For these, provide p values to 3 significant figures, as stated in the published policy (i.e. "for anything >0.001, please report to 3 significant figures, e.g. 0.00236 or 0.523, etc.").

In addition, Authors are asked to ensure clear annotation of data on right side of figure 4A, which shows two asterisks when three are referred to within the figure legend.

Comments to the Author:

At this time, initial review of your manuscript, "The apical mucus layer alters the pharmacological properties of the airway epithelium" is now complete. Attached herewith are detailed critiques of two expert Referees, as well as the Reviewing Editor's summary of their reviews.

Both Referees recognize the strong potential of your work to contribute important information to the field. However, specific questions warranting further attention also were raised. Both the Reviewing Editor and I feel that addressing these will increase your study's robustness. This encompasses further interrogation and analyses of alternative chloride pathways, additional clarification of protocols (especially pertaining to culture wash conditions), and scrutinizing potential effects of dithiothreitol on bioelectric properties of airway epithelial cultures. Please use the Referees's specific and detailed suggestions in guiding revision and in your response to their respective critiques.

We look forward to receiving your revised manuscript and thank you for your contribution to The Journal of Physiology.

REFeree COMMENTS

Referee #1:

This interesting study by Guidone et al. addresses the impact of the mucus layer covering the surface of primary airway epithelial cultures to modulate the response to CFTRinh-172 in Ussing chamber studies. The authors describe high variability of CFTRinh-172 response between cultures, where CFTRinh-172 only partially inhibited the cAMP-stimulated responses, whereas removal of mucus with a short DTT wash prior to experiments significantly enhanced potency of CFTRinh-172. Experiments using pro-inflammatory cytokine stimulations, which profoundly affect mucus composition and viscosity, showed that CFTRinh-172 had even lower potency, which was again increased by prior DTT wash. Similarly, experiments in CF cultures showed lower functional rescue by CFTR modulators, as estimated by CFTRinh-172 response, in cultures that were not washed vs. cultures with DTT wash. The authors conclude that the mucus layer impedes the potency of CFTRinh-172, which may lead to under-estimation of CFTR-mediated currents in Ussing chamber studies of primary airway epithelial cultures.

Strengths and weaknesses:

CFTRinh-172 is a specific and potent CFTR inhibitor that is commonly used in electrophysiology studies to quantify CFTR-dependent ion transport in primary airway epithelial cultures, therefore a highly important tool for the evaluation of therapeutic approaches and for promoting personalized medicine in cystic fibrosis. Therefore, understanding how the mucus layer affects CFTRinh-172 responses is of high importance. Previous reports on incomplete inhibition of cAMP-stimulated responses led to the assumption that other transporters/channels may contribute to cAMP-mediated chloride secretory response (e.g. SLC26A9). While the authors convincingly demonstrate that DTT washes increase CFTRinh-172 responses especially under conditions of increased mucus viscosity, there is little evidence provided, which would argue against the presence of an alternative chloride transporter. It is also not well understood, if more prolonged and regular PBS mucus

washes during differentiation (which is a common practice in laboratories working with ALI cultures) would be sufficient to remove adhesive mucus from the airway surface.

Specific comments:

1. The potency of CFTRinh-172 seems somewhat low compared to other reports (Saint-Criq et al., 2020). It is plausible that the authors' washing protocol (once every 3 days with PBS, no further details provided) could lead to accumulation of adhesive mucus. It is also not known if the cultures displayed cilia-dependent mucus transport (which typically develops several days after ciliation) at the time of cytokine treatments and end point experiments. Is it conceivable that these specific cell culture conditions used in this study favor the development of an adhesive mucus layer, which would explain the lower CFTRinh-172 sensitivity than what is found by others. Further, the authors used a lower, 10 μ M concentration of CFTRinh-172 (Saint-Criq et al., 2020 used 20 μ M). These limitations should be discussed.

2. The authors suggest that removal of mucus by DTT washes should be implemented as a routine procedure before Ussing chamber studies. However, DTT is a strong reducing agent and therefore can affect diverse cellular processes. To better understand the impact of DTT washes on the bioelectric properties of cultured epithelia, data on the change in I_{sc} should be provided for the responses to amiloride, cAMP-stimulation, CFTRinh-172, UTP and Ani9, as well as the residual current in presence/absence of DTT. Also, instead of I_{sc} ratios of pre- and post-CFTRinh-172, the authors should provide absolute changes in I_{sc}, at least in the online supplement

3. The authors suggest that these data argue against the presence of other chloride conductances than CFTR, however data is not provided on the residual current (i.e. remaining current after maximal cAMP stimulation and CFTR inhibition, which would indicate CFTR-independent chloride secretion) in presence/absence of DTT. To investigate residual chloride secretion, experiments should be performed at least with bumetanide applied in the end of the Ussing chamber protocol.

4. Figures 1,2,4: What does the dotted line designate on the figure?

5. Figure 1,3, 4: It is not clear what the individual data points represent in the figures. Are these individual filters or donors? Please clarify how many technical (filters) and biological (donors) replicates the data are based on.

Referee #2:

This is a simple and straightforward study examining the hindering effect of the epithelial mucin layer on access of CFTR inhibitors to the channel. The results are clear, and the information is quite useful technically and even clinically relevant (Figure 6). I have only a few minor comments:

1. Does stripping the mucus layer equally increase the efficacy of PPQ-102 and is the effect of PPQ-102 reversible as found in model systems?

2. I am not sure if this is feasible- but a very useful information can be to test if stripping the mucus increases the efficacy and the affinity of the CFTR correctors and potentiators. If the mucus-stripped tissue remains intact for 24 hrs, this can be tested and maybe used to refine treatment protocols for patients.

END OF COMMENTS

REPLY TO REVIEWING EDITOR AND REFEREES

Reviewing Editor:

The findings show that the mucus layer reduces CFTR_{inh}-172 efficacy, which may lead to underestimation of CFTR-mediated ion transport in experimental models if the layer is not removed. Given that CFTR inhibitors are widely used in electrophysiological studies for evaluating personalized CF treatment strategies, this is an important factor to consider.

We thank the Reviewing Editor for his/her positive comment.

To enhance the study's robustness the authors should

1) Include data on residual currents after maximal cAMP stimulation and CFTR inhibition, potentially with bumetanide, to clarify whether alternative chloride transport pathways contribute to the observed currents.

We thank the Reviewing Editor for his/her kind suggestion. All figures with short-circuit current recordings now include the absolute zero-current level indicated by a dashed line. In this way, it is possible to check the amount of current that remains after addition of chloride transport inhibitors, including bumetanide. Regarding bumetanide, we are now including experiments with this agent (shown Fig. 6).

2) Discuss how more frequent PBS washes during differentiation, as done in some laboratories, might impact mucus accumulation and CFTR_{inh}-172 efficacy.

We included in the Materials and Methods section a detailed description of the washing protocol during bronchial epithelial cells culture. We are also reporting the instructions provided by the PneumaCult ALI medium user manual. We are actually washing more extensively than suggested. However, in the Discussion section we are discussing the importance of the washing procedure.

3) Evaluate DTT Effects

We are now adding Fig. 7, in which DTT effects on electrogenic epithelial processes, i.e. ENaC, CFTR and TMEM16A currents are evaluated. Most of them are unaffected, with the exception of epithelia treated with IL-17A/TNF- α , in which ENaC activity was modestly but significantly increased. In the Discussion section, we are however discussing that the possible effect of DTT on other processes should be always considered when adopting the DTT procedure to remove mucus.

4) Add clarifications to Figures 1, 2, and 4, specifying what each data point represents (e.g., individual filters or donors) and the number of technical and biological replicates.

For all previous and new figures we are reporting now the number of experiments and of donors.

5) Test the impact of mucus stripping on the efficacy of other inhibitors like PPQ-102

We carried out short circuit current experiments with PPQ-102 instead of CFTR_{inh}-172 (Fig. 5). The results are comparable with those obtained with CFTR_{inh}-172: the mucus

stripping with DTT significantly increased the block of cAMP-activated current by PPQ-102.

Referee #1:

This interesting study by Guidone et al. addresses the impact of the mucus layer covering the surface of primary airway epithelial cultures to modulate the response to CFTRinh-172 in Ussing chamber studies. The authors describe high variability of CFTRinh-172 response between cultures, where CFTRinh-172 only partially inhibited the cAMP-stimulated responses, whereas removal of mucus with a short DTT wash prior to experiments significantly enhanced potency of CFTRinh-172. Experiments using pro-inflammatory cytokine stimulations, which profoundly affect mucus composition and viscosity, showed that CFTRinh-172 had even lower potency, which was again increased by prior DTT wash. Similarly, experiments in CF cultures showed lower functional rescue by CFTR modulators, as estimated by CFTRinh-172 response, in cultures that were not washed vs. cultures with DTT wash. The authors conclude that the mucus layer impedes the potency of CFTRinh-172, which may lead to under-estimation of CFTR-mediated currents in Ussing chamber studies of primary airway epithelial cultures.

CFTRinh-172 is a specific and potent CFTR inhibitor that is commonly used in electrophysiology studies to quantify CFTR-dependent ion transport in primary airway epithelial cultures, therefore a highly important tool for the evaluation of therapeutic approaches and for promoting personalized medicine in cystic fibrosis. Therefore, understanding how the mucus layer affects CFTRinh-172 responses is of high importance. Previous reports on incomplete inhibition of cAMP-stimulated responses led to the assumption that other transporters/channels may contribute to cAMP-mediated chloride secretory response (e.g. SLC26A9). While the authors convincingly demonstrate that DTT washes increase CFTRinh-172 responses especially under conditions of increased mucus viscosity, there is little evidence provided, which would argue against the presence of an alternative chloride transporter. It is also not well understood, if more prolonged and regular PBS mucus washes during differentiation (which is a common practice in laboratories working with ALI cultures) would be sufficient to remove adhesive mucus from the airway surface. We thank the referee for finding interesting our study and for his/her comments.

1. The potency of CFTRinh-172 seems somewhat low compared to other reports (Saint-Criq et al., 2020). It is plausible that the authors' washing protocol (once every 3 days with PBS, no further details provided) could lead to accumulation of adhesive mucus. It is also not known if the cultures displayed cilia-dependent mucus transport (which typically develops several days after ciliation) at the time of cytokine treatments and end point experiments. Is it conceivable that these specific cell culture conditions used in this study favor the development of an adhesive mucus layer, which would explain the lower

CFTRinh-172 sensitivity than what is found by others. Further, the authors used a lower, 10 μ M concentration of CFTRinh-172 (Saint-Criq et al., 2020 used 20 μ M). These limitations should be discussed.

We included in the Materials and Methods section a detailed description of the washing protocol during bronchial epithelial cells culture. The Pneumacult ALI medium user manual (now cited in the section) suggests to perform the washing procedure in week 2 post-airlift, approximately once per week, at room temperature and without any incubation. We improved this protocol with stronger washes, by starting them after the first week of ALI, incubating the PBS for 10 minutes at 37 °C and three times per week. We believe that this procedure reduces the formation of adhesive mucus. However, this improved washing procedure does not totally eliminate mucus as demonstrated by the immunofluorescence analysis and results with CFTR inhibitors. Therefore, DTT appears required. Regarding mucociliary clearance, epithelia under our culture conditions are highly ciliated and show mucociliary transport as shown in our previous study by microbead video-imaging (Guidone et al., JCI Insight 2022). Regarding the concentration of CFTR_{inh}-172, a concentration of 20 μ M could lead to a higher inhibition of CFTR currents. Actually, in preliminary experiments we used 20 μ M but the efficacy was still very limited in unwashed (no DTT) epithelia. We decided to use 10 μ M to stay away from the limit of solubility of CFTR_{inh}-172 which is close to 20 μ M. In the Discussion, we are now discussing the importance of the washing procedure to avoid mucus as much as possible.

2. The authors suggest that removal of mucus by DTT washes should be implemented as a routine procedure before Ussing chamber studies. However, DTT is a strong reducing agent and therefore can affect diverse cellular processes. To better understand the impact of DTT washes on the bioelectric properties of cultured epithelia, data on the change in I_{sc} should be provided for the responses to amiloride, cAMP-stimulation, CFTRinh-172, UTP and Ani9, as well as the residual current in presence/absence of DTT. Also, instead of I_{sc} ratios of pre- and post-CFTRinh-172, the authors should provide absolute changes in I_{sc}, at least in the online supplement

We thank the referee for this useful suggestion. We are now adding Fig. 7, in which DTT effects on electrogenic epithelial processes, i.e. ENaC, CFTR and TMEM16A currents are evaluated. Most of them are unaffected, with the exception of epithelia treated with IL-17A/TNF- α , in which ENaC activity was modestly but significantly increased. In the Discussion section, we are however discussing that the possible effect of DTT on other processes should be always considered when adopting the DTT procedure to remove mucus.

Regarding the residual current remaining after inhibition of chloride transport, we are now including in all short-circuit current recordings a dashed line indicating the absolute zero current level, which helps to intuitively estimate the absolute level of residual current after compounds addition. We think that this is a convenient way to indicate the extent of residual current remaining after treatments. It can be seen that in most cases the level reached by the current after CFTR inhibitor in DTT-washed epithelia is close to the absolute zero current level. Since there is variability in the absolute values between

epithelia, we think that the best way to show the difference in inhibitor efficacy is by current ratio.

3. The authors suggest that these data argue against the presence of other chloride conductances than CFTR, however data is not provided on the residual current (i.e. remaining current after maximal cAMP stimulation and CFTR inhibition, which would indicate CFTR-independent chloride secretion) in presence/absence of DTT. To investigate residual chloride secretion, experiments should be performed at least with bumetanide applied in the end of the Ussing chamber protocol.

We thank the referee for his/her suggestions. We have now included experiments with bumetanide added after CFTR_{inh}-172 (Fig. 6). Interestingly, the relative effects of CFTR_{inh}-172 and bumetanide are inversely related. In DTT-washed epithelia, the relative effect of CFTR_{inh}-172 is increased and that of bumetanide is decreased, as expected if the two agents act on the same pathway (i.e. CFTR-dependent chloride transport). Also, the total effect of CFTR_{inh}-172 plus bumetanide was close to full inhibition of the cAMP-activated current in epithelia kept under control conditions or treated with IL-4 (Fig. 6A and B). Instead, a substantial fraction of the current remained unblocked in epithelia treated with IL-17A/TNF- α (Fig. 6C). A possible explanation is that, besides chloride secretion, these epithelia have an increased bicarbonate secretion, as already reported by others (Zajac et al., Front Pharmacol 2023). We are now considering this issue and commenting the results in the Discussion section.

4. Figures 1,2,4: What does the dotted line designate on the figure?

The dotted line indicates the baseline current that was considered to measure the cAMP-activated current. This is now explained in the legends to figures. As specified above, we have now also included dashed line that indicates the absolute zero current level.

5. Figure 1,3, 4: It is not clear what the individual data points represent in the figures. Are these individual filters or donors? Please clarify how many technical (filters) and biological (donors) replicates the data are based on.

We have now specified in each legend the number of filters and donors.

Referee #2:

This is a simple and straightforward study examining the hindering effect of the epithelial mucin layer on access of CFTR inhibitors to the channel. The results are clear, and the information is quite useful technically and even clinically relevant (Figure 6).

We thank the referee for his/her positive comment.

I have only a few minor comments:

1. Does stripping the mucus layer equally increase the efficacy of PPQ-102 and is the effect of PPQ-102 reversible as found in model systems?

We thank the referee for his/her suggestion. We have now included experiments with

PPQ-102 (Fig. 5). The results are similar to those obtained with CFTR_{inh}-172: the mucus stripping with DTT significantly increased the efficacy of PPQ-102.

2. I am not sure if this is feasible- but a very useful information can be to test if stripping the mucus increases the efficacy and the affinity of the CFTR correctors and potentiators. If the mucus-stripped tissue remains intact for 24 hrs, this can be tested and maybe used to refine treatment protocols for patients.

We have not tested the effect of mucus on CFTR rescue because we usually apply correctors only from the basolateral side.

Dear Dr Galietta,

Re: JP-RP-2025-287891R1 "THE APICAL MUCUS LAYER ALTERS THE PHARMACOLOGICAL PROPERTIES OF THE AIRWAY EPITHELIUM" by Daniela Guidone, Martina De Santis, Emanuela Pesce, Valeria Capurro, Nicoletta Pedemonte, and Luis J.V. Galietta

Thank you for submitting your manuscript to The Journal of Physiology. It has been assessed by a Reviewing Editor and by 2 expert referees and we are pleased to tell you that it is acceptable for publication following satisfactory revision.

REVISION CHECKLIST:

We look forward to receiving your revised submission.

Yours sincerely,

Peying Fong
Senior Editor
The Journal of Physiology

REQUIRED ITEMS

- You must start the Methods section with a paragraph headed Ethical Approval. If experiments were conducted on humans, confirmation that informed consent was obtained, preferably in writing, that the studies conformed to the standards set by the latest revision of the Declaration of Helsinki and that the procedures were approved by a properly constituted ethics committee, which should be named, must be included in the article file. If the research study was registered (clause 35 of the Declaration of Helsinki), the registration database should be indicated, otherwise the lack of registration should be noted as an exception (e.g. The study conformed to the standards set by the Declaration of Helsinki, except for registration in a database). For further information see: <https://physoc.onlinelibrary.wiley.com/hub/human-experiments>.

PLEASE NOTE: IT IS THE 'CLAUSE 35' STATEMENT THAT IS CURRENTLY MISSING. PLEASE CONFIRM WHETHER YOUR STUDY WAS A CLINICAL TRIAL, OR NOT.

If not, please update your Methods to read: 'The study conformed to the standards set by the latest revision of the Declaration of Helsinki, except for registration in a database'.

EDITOR COMMENTS

Reviewing Editor:

Please address Referee #1's comment regarding the discussion of bumetanide experiments, specifically omitting the statements about the lack of other chloride conductances.

Please also see 'Required Items' above.

Senior Editor:

The review of your revised manuscript, "THE APICAL MUCUS LAYER ALTERS THE PHARMACOLOGICAL PROPERTIES OF THE AIRWAY EPITHELIUM" is now complete. Both Referees and the Reviewing Editor concur on its suitability for acceptance. At this stage, both Referees are largely satisfied with how you have chosen to address comments raised in initial review. Note that Referee 1 does ask that you make one minor change by tempering phrasing within the Key Findings. Both the Reviewing Editor and I expect that you will be able to perform this edit readily.

Thank you for contributing your work to The Journal of Physiology. I look forward to receiving what I expect will be a final revision.

REFEREE COMMENTS

Referee #1:

I thank the authors for providing additional data and clarifications, and presenting a more balanced discussion of the data. I have one remaining issue with the discussion of the bumetanide experiments. The authors convincingly show that adding bumetanide in the end of the experiment further reduces the current in untreated cultures, which in case of DTT washed

cultures, appears to be ~20% of the total current (Figure 6A). This would clearly argue for the presence of other chloride conductance, which cannot be inhibited by CFTRinh172. Therefore, I suggest to omit the statements regarding the lack of other chloride conductances in the key finding summary: "The partial effect of CFTRinh-172 inhibitors is due to the presence of the mucus layer and not to non-CFTR channels/transporters" as well as in the discussion "Therefore, partial efficacy of CFTRinh-172 CFTR inhibitors in blocking cAMP-activated anion transport is probably due to mucus and not to the contribution of other ion channels/transporters."

Referee #2:

No further comments.

END OF COMMENTS

Answer to reviewer request

We have toned down the last sentence of the key point list to address reviewer request: we have removed the reference to other channels and transporters.

Also, we have deleted the sentence on the exclusion of other channels and transporters at the end of discussion, as requested.

Dear Associate Professor Galletta,

Re: JP-RP-2025-287891R2 "THE APICAL MUCUS LAYER ALTERS THE PHARMACOLOGICAL PROPERTIES OF THE AIRWAY EPITHELIUM" by Daniela Guidone, Martina De Santis, Emanuela Pesce, Valeria Capurro, Nicoletta Pedemonte, and Luis J.V. Galletta

We are pleased to tell you that your paper has been accepted for publication in The Journal of Physiology.

Yours sincerely,

Peying Fong
Senior Editor
The Journal of Physiology

If you would like to receive our 'Research Roundup', a monthly newsletter highlighting the cutting-edge research published in The Physiological Society's family of journals (The Journal of Physiology, Experimental Physiology, Physiological Reports, The Journal of Nutritional Physiology and The Journal of Precision Medicine: Health and Disease), please click this link, fill in your name and email address and select 'Research Roundup':
<https://www.physoc.org/journals-and-media/membernews>

- You can help your research get the attention it deserves! Check out Wiley's free Promotion Guide for best-practice recommendations for promoting your work at: www.wileyauthors.com/eeo/guide. You can learn more about Wiley Editing Services which offers professional video, design, and writing services to create shareable video abstracts, infographics, conference posters, lay summaries, and research news stories for your research at: www.wileyauthors.com/eeo/promotion.

EDITOR COMMENTS

Reviewing Editor:

Comments to the Author:
CONGRATULATIONS!

Senior Editor:

Comments to the Author:

Thank you for your responsiveness in addressing the final details identified in the last round of review. I am pleased you chose to contribute this manuscript *The Journal of Physiology*, and congratulate you on an interesting and important study.